# Comparison of Mid-Infrared Handheld and Benchtop Spectrometers to Detect *Staphylococcus epidermidis* in Bone Grafts

**DOI:** 10.3390/bioengineering10091018

**Published:** 2023-08-29

**Authors:** Richard Lindtner, Alexander Wurm, Katrin Kugel, Julia Kühn, David Putzer, Rohit Arora, Débora Cristina Coraça-Huber, Philipp Zelger, Michael Schirmer, Jovan Badzoka, Christoph Kappacher, Christian Wolfgang Huck, Johannes Dominikus Pallua

**Affiliations:** 1Department of Orthopaedics and Traumatology, Medical University of Innsbruck, Anichstraße 35, 6020 Innsbruck, Austria; richard.lindtner@i-med.ac.at (R.L.); katrin-kugel@gmx.de (K.K.); julia.kuehn@student.i-med.ac.at (J.K.); david.putzer@i-med.ac.at (D.P.); rohit.arora@tirol-kliniken.at (R.A.); debora.coraca-huber@i-med.ac.at (D.C.C.-H.); johannes.pallua@i-med.ac.at (J.D.P.); 2Praxis Dr. Med. Univ. Alexander Wurm FA für Orthopädie und Traumatologie, Koflerweg 7, 6275 Stumm, Austria; 3University Clinic for Hearing, Voice and Speech Disorders, Medical University of Innsbruck, Anichstraße 35, 6020 Innsbruck, Austria; philipp.zelger@tirol-kliniken.at; 4Department of Internal Medicine, Clinic II, Medical University of Innsbruck, Anichstraße 35, 6020 Innsbruck, Austria; schirmer.michael@icloud.com; 5Institute of Analytical Chemistry and Radiochemistry, University of Innsbruck, Innrain 80-82, 6020 Innsbruck, Austria; jbadzoka@gmail.com (J.B.); christoph.kappacher@uibk.ac.at (C.K.); christian.w.huck@uibk.ac.at (C.W.H.)

**Keywords:** bone quality, handheld FTIR spectrometer, attenuated total reflectance spectroscopy, principal component analyses, *Staphylococcus epidermidis*

## Abstract

Bone analyses using mid-infrared spectroscopy are gaining popularity, especially with handheld spectrometers that enable on-site testing as long as the data quality meets standards. In order to diagnose *Staphylococcus epidermidis* in human bone grafts, this study was carried out to compare the effectiveness of the Agilent 4300 Handheld Fourier-transform infrared with the Perkin Elmer Spectrum 100 attenuated-total-reflectance infrared spectroscopy benchtop instrument. The study analyzed 40 non-infected and 10 infected human bone samples with *Staphylococcus epidermidis*, collecting reflectance data between 650 cm^−1^ and 4000 cm^−1^, with a spectral resolution of 2 cm^−1^ (Agilent 4300 Handheld) and 0.5 cm^−1^ (Perkin Elmer Spectrum 100). The acquired spectral information was used for spectral and unsupervised classification, such as a principal component analysis. Both methods yielded significant results when using the recommended settings and data analysis strategies, detecting a loss in bone quality due to the infection. MIR spectroscopy provides a valuable diagnostic tool when there is a tissue shortage and time is of the essence. However, it is essential to conduct further research with larger sample sizes to verify its pros and cons thoroughly.

## 1. Introduction

Bone represents the second most commonly transplanted tissue behind blood [1]. In orthopedic surgery, the application of human bone allografts is prevalent in promoting spinal fusion and reconstructing bone defects resulting from trauma, tumors, or revision arthroplasty [2,3,4,5,6,7,8]. Removing loose implants and adjacent fibro cellular membranes is often necessary in the context of hip arthroplasty revision procedures. To address any resulting bone loss, particulate bone grafts are commonly employed as a compensatory measure. In order to induce bone remodeling and prevent early implant subsidence, the morselized allograft must be compacted [6]. Upon attaining initial stability via impaction procedures, the graft may be assimilated into the host skeleton through revascularization [9,10,11,12]. According to soil mechanics, firmly compacted aggregates should have well-graded particle sizes and be rigidly contained [13]. Bone grafts can be employed using tissue from the same patient (autografts) or the same species (allografts) [14]. These grafts are typically harvested during surgical procedures and preserved at −80 °C for up to five years. Before use, bone banks conduct rigorous screening of the graft material to ensure the recipient’s safety from potential infectious pathogens [6]. Moreover, legislative standards for quality controls addressing contamination and communicable illnesses have been created. It is highly recommended that bone transplants be washed and treated with antibiotics before being stored at a temperature of −80 °C to guarantee their safety and effectiveness. Aside from offering mechanical support, bone grafts can also act as a local source of antibiotic treatment, which can help prevent and treat any possible infections in the vicinity. This approach has been proven effective and is widely accepted as a standard practice [5,15,16]. Clinicians, patients, and healthcare providers face a significant challenge with orthopedic-implant-related infections, including infections in joints and fractures. These infections can lead to high rates of disability and death, impact quality of life, and place a significant financial burden on the healthcare system [17,18]. The orthopedic community is increasingly concerned about controlling implant-associated infections due to the projected increase in patients suffering from this complication [19]. Treating these infections can be challenging and often requires surgical procedures [20,21]. It is worth noting that periprosthetic joint infections (PJIs) can be quite severe, with fatality rates that are comparable to those of breast cancer and melanoma [8]. The most commonly found microbes in bone and implant-related infections are Coagulase-Negative Staphylococci (CoNS), mainly *Staphylococcus epidermidis*. They are followed by *Staphylococcus aureus* and mixed flora [14,22,23]. Over time, CoNS and other microbes have become increasingly resistant to commonly used antibiotics such as penicillin, oxacillin, ciprofloxacin, clindamycin, erythromycin, and gentamicin, significantly reducing the effectiveness of these drugs [24]. The capacity of these bacteria to form biofilms is another factor contributing to this phenomenon and adversely influencing CoNS’s antimicrobial susceptibility [25]. Moreover, biofilm development explains why some common flora species historically regarded as “harmless” turn infectious when they settle on the surface of foreign objects. Detecting colonization bacteria and searching for concealed biofilms on these allografts is crucial to preventing contamination and biofilm development during bone grafting [26,27,28,29,30,31,32,33]. Over the years, it has been demonstrated that laboratory spectroscopy in the mid-infrared (MIR) may be used in diagnosing various bone diseases in humans [34,35,36,37,38,39,40,41,42] and bone quality [43]. MIR spectroscopy employs infrared radiation to engage with molecular vibrations, covering the region of the electromagnetic spectrum from 4000 to 400 cm^−1^. In this region, infrared radiation can excite molecular vibrations, such as stretching, bending, and twisting modes, leading to changes in the molecule’s dipole moment. The absorption of infrared radiation by a sample leads to the formation of distinct absorption bands, which can be effectively exploited to identify and quantify the molecular constituents of the sample [44,45].

In recent years, handheld spectroscopy has gained more attention alongside laboratory spectroscopy, as it aims to quickly and accurately characterize on-site materials. This method encompasses a range of techniques that are highly effective in their applications. Thus, handheld MIR spectrometers have several advantages over traditional benchtop spectrometers. They are lightweight, compact, and battery-powered, enabling on-site analysis in various fields and requiring minimal sample preparation, providing rapid results [46]. Furthermore, they are relatively affordable compared to benchtop spectrometers. However, they also have some limitations. They typically have lower spectral resolution than benchtop spectrometers, limiting their ability to distinguish between closely spaced absorption bands. Moreover, they have a lower signal-to-noise ratio, which can lead to inaccurate results. In addition, handheld MIR spectrometers have a limited spectral range, which can limit their ability to analyze complex samples [47]. Achieving the best bone graft samples for patients involves performing a spectroscopic quality test. This test can effectively determine the bone’s typical composition, including phosphate, carbonate bone mineral, collagen, and contaminants such as *Staphylococcus epidermidis*. Experts then must confidently identify and select only the highest-quality bone grafts through handheld MIR spectroscopy. This tool provides crucial information about mineralization processes and can efficiently detect contamination [42,48,49]. In the last ten years, several studies have highlighted the usefulness of MIR spectroscopy for detecting different bone diseases in humans [34,35,36,37,38,39,40,41], including the evaluation of bone quality [43]. Bone comprises a variety of MIR bands, typically composed of phosphate (ν_3_PO_4_^3−^), carbonate (ν_1_CO_3_^2−^), collagen matrix, amide III, CH_2_ of protein, and amide I [41,50,51,52,53]. *Staphylococcus epidermidis* is the most prevalent pathogen in bone and implant-related infections. Therefore, this study aimed to compare the efficacy of the small handheld instrument with a benchtop mid-infrared spectrometer in detecting *Staphylococcus epidermidis* on bone grafts. The small Agilent 4300 Handheld Fourier-transform infrared (FTIR) scanner and the larger Perkin Elmer Spectrum 100 attenuated-total-reflectance infrared spectroscopy (ATR-IR) benchtop instrument were used, and a principal component analysis was conducted on fresh frozen bone samples. This study compares mid-infrared handheld and benchtop spectrometers to detect *Staphylococcus epidermidis* in bone grafts. The outcome of this study provides new insights into the potential of small MIR handheld spectrometers in identifying these pathogens in human bone grafts.

## 2. Materials and Methods

### 2.1. Sample Collection

The femoral heads used in our bone bank were sourced from individuals who had undergone hip replacement surgery due to advanced hip osteoarthritis or femoral neck fracture. Before their donation, all patients provided written informed consent, ensuring that their contribution was made willingly and with a full understanding of the process. The donated bone that did not fulfil the criteria for therapeutic use (e.g., due to incomplete screening and documentation) was kept at the bone bank and utilized in scientific studies. In general, severe osteoporosis and contaminated samples caused by various pathogens were eliminated and not collected, regardless of the research. There were no criteria based on age or gender. The bone was drained and chilled with 0.9% saline during osteotomy to prevent heat damage. Cartilage and cortical tissues from the femoral heads were removed using a bone saw. Bone chips 3–5 mm in diameter were extracted from the residual spongious tissue (Noviumagus Bone Mill; Spierings Meische Techniek BV, Nijmegen, The Netherlands) with a bone mill. A total of 40 human bone samples were examined. The local ethics council approved the retrospective study (EK 1291/2021) per the guiding principles outlined in the Declaration of Helsinki.

### 2.2. Development of Biofilm on Bone Allografts

To examine *Staphylococcus epidermidis* ATCC 12228, Mueller–Hinton broth with confidence and precision was utilized. We carefully incubated the broth at 37 °C for 24 h. The inoculum was diluted to 106 CFU/mL, and 200 µL of the suspension was added to each well of a multi-well plate. Individual fresh frozen bone allografts and a substrate were inserted to create biofilms. The plates were then placed in an orbital shaker and incubated at 37 °C for 48 h to form the biofilms. Once completed, the supernatant was removed, and the bone samples were washed with new PBS to ensure the removal of any planktonic bacteria. After this procedure, the bone samples were dried in an aspirator (3.2 kPa) for 10 min at room temperature and measured. The drying time was sufficient, as prolonging the drying time to 24 h caused no differences in spectra quality.

### 2.3. Benchtop Perkin Elmer Spectrum 100 ATR-IR Spectrometer

The MIR ATR-IR spectra were collected using a Perkin Elmer Spectrum 100 ATR-IR spectrometer (Perkin Elmer, Waltham, MA, USA). There were eight scans per sample from three positions, with a resolution of 0.5 cm^−1^ and a wavenumber range of 4000 to 650 cm^−1^. The measurement was conducted at a temperature of 22 °C under controlled humidity levels.

### 2.4. Agilent 4300 Handheld FTIR

MIR spectra were obtained using the Agilent 4300 Handheld FTIR (Agilent Technologies, Santa Clara, CA, USA) device. The spectral range covered was from 4000 to 650 cm^−1^, with a spectral resolution of 2 cm^−1^. Each sample was subjected to eight spectra recorded from three different positions. The measurement was conducted at a temperature of 22 °C under controlled humidity levels.

### 2.5. Data Processing

Data processing was performed using the Spectrum software version 6.3.1.0134 (Perkin Elmer, Waltham, MA, USA) and the Unscrambler X 10.5 (AspenTech, Bedford, MA, USA).

Spectral Parameters: The diagnostic parameters were studied using peak intensity (I) [54,55,56,57,58,59]. The determination of intensities was made effortless through an Excel spreadsheet that handles spectroscopic and chromatographic data [60]. A statistical analysis of the spectral parameters was conducted using GraphPad Prism software (version 9, San Diego, CA, USA) and compared through a two-sample *t*-test. A significant result is only considered if the *p*-value is less than 0.05.

### 2.6. Principal Component Analyses (PCA)

PCA models were created using Unscrambler X 10.5, which involved importing spectra into the program and applying various data pretreatments, such as a reduction factor 36, Savitzky–Golay smoothing with 15 smoothing points, and area normalization.

## 3. Results

The main objective of this study was to compare the effectiveness of MIR handheld and benchtop spectrometers in detecting *Staphylococcus epidermidis* in bone grafts. Of 40 non-infected human bone samples, 10 were intentionally infected with *Staphylococcus epidermidis* ATCC 12228. Twenty-two females and eighteen males provided non-infected human bones (*n* = 40), while the infectious human bones were collected from seven females and three males (*n* = 70/30%). The sample characteristics are summarized in Table 1. Figure 1 shows the advantages and disadvantages of conventional infection diagnosis, MIR handheld, and benchtop spectrometry. Compared to MIR, the traditional diagnosis of infection can be time-consuming, resource-intensive, and labor-intensive.

### 3.1. Spectroscopy Data Evaluation

When examining non-infected human bones, certain MIR bands are usually observed. These bands include phosphate (ν_3_PO_4_^3−^), carbonate (ν_1_CO_3_^2−^), collagen matrix, amide III, CH_2_ of protein, and amide I [41,50,51,52,53,61,62,63]. The phosphate of human bone exhibits four distinct internal vibration modes. These modes comprise ν_1_ (~960 to 961 cm^−1^), ν_2_ (~420 to 450 cm^−1^), ν_3_ (~1035 to 1048 cm^−1^, and ~1070 to 1075 cm^−1^), and ν_4_ (~587 to 604 cm^−1^) [61,62,64,65]. Organic components, on the other hand, exhibit bands within the range from ~1200 to 1320 cm^−1^ (amide III), ~1595 to 1700 cm^−1^ (amide I), ~1400 to 1470 cm^−1^, and ~2800–3000 cm^−1^ (C-H groups) [62,64]. The collagen matrix is assigned to the peaks at 851, 873, and 917 cm^−1^, respectively, whereas phenylalanine is characteristic of the peak at 1001 cm^−1^ [41,51,52]. Furthermore, the peak at 1450 cm^−1^ corresponds to CH_2_ deformation [58]. Finally, the CO_3_^2−^ carbonate group’s internal modes are detected at 1070 cm^−1^ (B-type carbonate) and 1103 cm^−1^ (A-type carbonate) [62]. Type-B carbonate (substitutions of phosphate) and type-A carbonate (substitutions of hydroxide) significantly impact the properties of apatite crystals, including their perfection and crystallite domain size. These effects have important implications for material properties in different environmental conditions. It is worth noting that type-B carbonate is dominant in forming apatitic biominerals, while type-A carbonate controls the properties of apatitic minerals in non-physiological states [66,67,68]. Figure 2 illustrates a representative spectrum of a non-infected human bone, using the Agilent 4300 Handheld and Perkin Elmer Spectrum 100 instruments. While the basic shape of the mean spectra was similar between different measurements for both devices, the recorded MIR spectra exhibited varying absorbance values. Specifically, the Perkin Elmer Spectrum 100 instrument had spectra ranging from 0 ± 0 to 0.9555 ± 0.01976, while the Agilent 4300 Handheld showed minimum-to-maximum differences from 0.0004626 ± 7.614 × 10^−5^ to 0.0007611 ± 5.644 × 10^−5^. Notably, the positions and shape of the v_3_PO_4_^3−^ band from 1200 to 900 cm^−1^, the amide I band from 1730 to 1585 cm^−1^, and the bending and stretching modes of C-H groups from ~3000 to 2800 cm^−1^ coincided nicely. However, there were some differences in the Agilent 4300 Handheld, particularly in the shape of the feature between 650 and 1800 cm^−1^ (i.e., v_3_CO_3_^2−^ and amide III), and the amide II feature at around 1650 cm^−1^ was not displayed in the Agilent 4300 Handheld as it was in the Perkin Elmer Spectrum 100 instrument.

Figure 3 clearly compares the average spectra of non-infected and infected human bone with *Staphylococcus epidermidis*. This was achieved through measurements taken using the Agilent 4300 Handheld and Perkin Elmer Spectrum 100 instruments. The average spectra were derived from eight spectra by using the Perkin Elmer Spectrum 100 and four by using the Agilent 4300 Handheld. Notably, infected and non-infected human bones exhibit distinct spectral features, including the v3PO_4_^3−^ band, the amide I band, and C-H groups from ~3000 to 2800 cm^−1^. However, significant differences between infected and non-infected human bones are present in the fingerprint region from 1800 to 650 cm^−1^, the v_3_PO_4_^3−^ band, and the amide I band. Specifically, the Agilent 4300 Handheld instrument demonstrates a substantial loss of the amide I band in infected human bones.

Table 2 concisely overviews the essential parameters scrutinized in MIR spectra. These parameters encompass the intensities (I) of the most influential bands, culminating in single-band and multiple-band ratios [54,55,57,58,59]. They serve as markers that furnish precise diagnostic data on infections that impact human bones, as depicted in Figure 4.

The results from a two-sample *t*-test, displayed in Figure 4, indicate a significant difference in the levels of phosphate, amide I, and CH-Aliphatic content (CHACont) between infected and non-infected samples, using the Agilent 4300 Handheld instrument. The Perkin Elmer Spectrum 100 instrument revealed a significant difference in mineral/matrix (MMR), mineral quality and crystallinity, and mineral carbonate content (MinCarb) between infected and non-infected cases. Additionally, bone quality and strength can be assessed through the mineral-to-matrix ratio (MMR), which measures mineralization levels [41,42,69,70]. Calculating the ratio of mineral-specific MIR band intensities (phosphate and carbonate bands) to the intensity of the amide I band or the ratio of phosphate band intensity to the total intensity of proline and hydroxyproline MIR bands is crucial in determining changes in bone strength. It is important to note that bacterial infection can result in a higher loss of relative mineral content in bones. This makes weaker bones more susceptible to fractures than non-infected human bones [71,72]. The ratio of carbonate to phosphate intensities in MMR indicates certain properties.

Additionally, the amide I band can help analyze changes in the collagen network caused by infection, as it is a typical protein conformation indicator due to its role in cross-linking and bonding. This band serves as an indicator of the protein structure [42]. Figure 4 clearly demonstrates that bacterial infection in human bones decreases structural organization and relative collagen. To measure the organic and inorganic components within the bone, the CH-aliphatic content (CHACont) is measured, which is typically attributed to the presence of proteins and lipids [73]. The study results reveal that bacterial infection causes a significantly higher loss of CHACont in bones. Co-culturing *Staphylococcus epidermidis* with human bone samples results in a significant deterioration of both bone quality and protein conformation. PCA was conducted to thoroughly describe the full range of spectral variations, as it is impossible to identify exact correlations with this type of processing.

### 3.2. Diagnostic Performance PCA

With the aim of facilitating a prompt diagnosis of the causative agent of infection, molecular techniques are commonly used in routine diagnostics. Early detection of infections is critical for guiding treatment decisions and improving patient outcomes [74]. The study successfully utilized the Agilent 4300 Handheld and Perkin Elmer Spectrum 100 instruments in the MIR with PCA analysis to expertly distinguish between bone graft samples infected with *Staphylococcus epidermidis* and those not infected. The potential diagnostic utility of PCA analysis of spectroscopic data has been demonstrated in previous studies [42,75,76,77,78,79,80,81,82,83]. PCA was utilized on the averaged spectra of Agilent 4300 Handheld and Perkin Elmer Spectrum 100 instruments to conduct the analysis. Our study involved examining 40 non-infected and 10 infected bone samples, following the methodology outlined in previous studies [42,83,84]. Figure 5 presents the results of the spectral analyses conducted using PCA. It compares non-infected and infected human bone samples through a score plot of the first and second principal components. Table 3 provides a detailed analysis of five wavenumber ranges used in PCA.

According to the PCA models, the most informative data are located in regions II, IV, and V, as displayed in Figure 5. The score plots demonstrate the correlation between PC1 and PC2 for non-infected and infected bone samples within the II, IV, and V ranges, along with the corresponding loadings of PC1. The red symbols signify non-infected samples, while the blue symbols represent infected samples. PC1 accounts for 95%, 100%, and 91% for Perkin Elmer Spectrum 100 instrument and 97%, 98%, and 91% for Agilent 4300 Handheld in the spectral regions of amide III, amide I, and bending and stretching modes of C-H groups, as depicted in Figure 5 II, IV, and V Both methods are trustworthy and robust and can be enhanced to discriminate *Staphylococcus epidermidis*, *Staphylococcus aureus*, and other bacteria. PCA analysis is automated and objective, making it a beneficial tool for the routine laboratory screening of bone samples for bacterial infections. This method can significantly improve diagnostic performance for laboratories not specifically detecting bacterial bone infections.

## 4. Discussion

Both the Agilent 4300 Handheld instrument and the Perkin Elmer Spectrum 100 are trustworthy and robust and can be enhanced to discriminate *Staphylococcus epidermidis*, *Staphylococcus aureus*, and other bacteria. However, different wavelengths are responsible for the comparable outcome of the two instruments: For non-infected bones, the Agilent 4300 Handheld and the Perkin Elmer Spectrum 100 instrument equally detect phosphate, amide I (region IV), and stretching modes of CH groups (region V) with different intensities, especially for phosphate (Figure 2). The Agilent 4300 Handheld, however, has a lower resolution for ν_3_CO_3_^2−^ than the Perkin Elmer Spectrum 100 instrument. Thus, bone quality and strength cannot be assessed equally through the mineral-to-matrix ratio (MMR), which measures mineralization levels [41,42,69,70]. Additionally, mineral quality, crystallinity, and mineral carbonate content (MinCarb) cannot be assessed equally. Concerning bones infected with Staphylococcus epidermidis, the Agilent 4300 Handheld instrument especially differentiates the levels of phosphate, amide I (region IV), and CH-Aliphatic content (CHACont, region V), whereas the Perkin Elmer Spectrum 100 instrument revealed a significant difference in amide I (region IV), mineral/matrix (MMR), mineral quality and crystallinity, and mineral carbonate content (MinCarb) between infected and non-infected bones. In summary, according to the PCA models, the most informative data are located in regions II, IV, and V (Figure 5). This shows that the Agilent 4300 Handheld instrument does not perform well between 650 and 1800 cm^−1^, whereas the result of amide I (region IV) especially shows comparable data concerning the spectral analysis and PCA. The correlation between PC1 and PC2 was examined for non-infected and infected bone samples in the II, IV, and V ranges. The loadings of PC1 were also analyzed, with PC1 accounting for 97%, 98%, and 91% for the Agilent 4300 Handheld and 95%, 100%, and 91% for the Perkin Elmer Spectrum 100 instrument in the spectral regions of II (amide III), IV (amide I), and V (bending and stretching modes of C-H groups) (as shown in Figure 5).

Further investigations utilizing the carbonate/phosphate ratio reveal that human bone samples co-cultured with *Staphylococcus epidermidis* exhibit a significant reduction in bone quality and protein conformation. Infected bones, in particular, demonstrate a more pronounced decrease in relative mineral content compared to non-infected bones. Additionally, changes in the collagen network can be identified through the amide I band. PCA allows us to detect *Staphylococcus epidermidis* in multiple spectral regions, primarily from amide III, amide I, and C-H groups’ bending and stretching modes, thereby validating its presence. These results using MIR spectroscopy are in line with previously published Raman investigations [41,42]. Such a direct detection of bone infection via MIR or Raman spectroscopy is challenging due to the complexity of bone tissue, low pathogen concentrations, and spectral overlaps. Nevertheless, with the integration of complementary techniques and careful optimization of sample preparation and data analysis, there is potential for MIR spectroscopy to contribute to the laser-independent detection and characterization of bone infections.

MIR spectroscopy is a speedy, robust, and automatable technique with significant advantages. MIR spectroscopy also requires a tiny untreated bone sample, making it suitable for situations with limited bone availability. Finally, this technique may benefit patients requiring urgent treatment because the analysis results are obtained without delay. In contrast, intraoperative tissue cultures, which are the current gold standard in diagnosing periprosthetic joint infections, are resource-intensive, and results can be expected only 5 to 11 days after tissue samples have been intraoperatively obtained [85]. Similarly, a histopathological workup of tissue samples requires significant resources and time, while sensitivity and specificity are even lower [86,87]. Another issue of tissue cultures is that inadequate processing and transport to the laboratory may result in either false-positive results due to contamination or culture-negative infections [88]. Overall, the sensitivity and specificity of intraoperative tissue cultures to detect periprosthetic joint infection have been reported to range from 0.51 to 0.90 and from 0.67 to 1.00, respectively, and diagnostic accuracy may be increased by additional sonication fluid cultures from explanted prosthetic components [89]. Similar values have been reported for spinal implant infections [90]. However, antibiotic therapy before culture sampling further decreases the sensitivity of intraoperative tissue cultures and results in culture negativity in a significant portion of periprosthetic joint infections [91]. MIR spectroscopy not only appears to help overcome the suboptimal results and well-known limitations of current infection diagnosis but may also contribute to further reducing the risk of transmitting bacterial contamination by bone allografts.

Our research has several limitations, including the small sample size and the prespecified incubation time of 48 h for bone fragments to promote the bacterial growth of *Staphylococcus epidemridis*. Therefore, the pathogen’s limit of detection (LOD) could not be determined. Further studies to achieve calibration standards with known pathogen concentrations assisting in LOD determination are warranted. Our findings highlight the potential of spectroscopic analyses to link molecular changes with pathological conditions. Based on these findings, we anticipate that handheld and benchtop MIR spectroscopy have a different potential and will be used to detect bacterial infections in human bone samples either in biobanks or under surgical conditions, thus adding rapid information to other microbial diagnostic procedures. Further research with larger sample sizes and various incubation times are required to control for potential confounding variables and to validate this technique as a new diagnostic tool for the clinical handling of bones, combining MIR data and machine-learning analysis.

## 5. Conclusions

These findings highlight the comparable potential of the Agilent 4300 Handheld and Perkin Elmer Spectrum 100 for detecting infections in bone grafts. The data are consistent with previous research that suggests MIR spectrometry’s effectiveness in identifying bacterial infections and extend the work now even for handheld instruments.

The Agilent 4300 Handheld and Perkin Elmer Spectrum 100 Benchtop spectrometers have unique strengths and limitations. The handheld spectrometer is beneficial for on-site and immediate clinical use, while the benchtop spectrometer is more suitable for research and complex analytical tasks in a laboratory setting. Thus, the choice between the two depends on specific requirements, portability needs and the budget as pros of the Agilent 4300 Handheld instrument, and the resources needed as con for the Perkin Elmer Spectrum 100 instrument. Using a handheld instrument has broad implications for the medical community, as such instruments allow MIR spectrometry to effectively and efficiently detect infections in bone grafts, potentially reducing healthcare costs and improving patient outcomes. Researchers and analysts should carefully consider these factors when selecting the most appropriate spectrometer for bacterial detection in bone grafts.

In summary, Agilent 4300 Handheld and Perkin Elmer Spectrum 100 instruments are both promising tools to diagnose human bone infections in situations with limited tissue availability and high urgency for treatment. However, further assessment is necessary to further validate the technique’s benefits and drawbacks with a larger sample size and for more microbes and fungi.

## Figures and Tables

**Figure 1 bioengineering-10-01018-f001:**
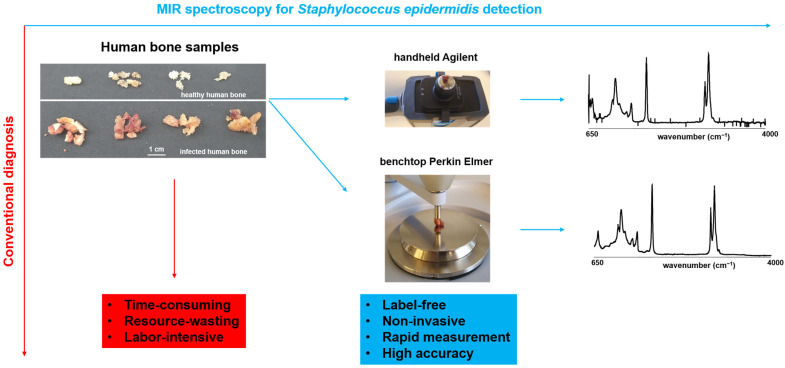
Advantages and disadvantages of MIR spectroscopy and conventional infection diagnosis. Comparison of the MIR devices used in the experiment (Agilent handheld device and Perkin Elmer benchtop) and resulting spectra.

**Figure 2 bioengineering-10-01018-f002:**
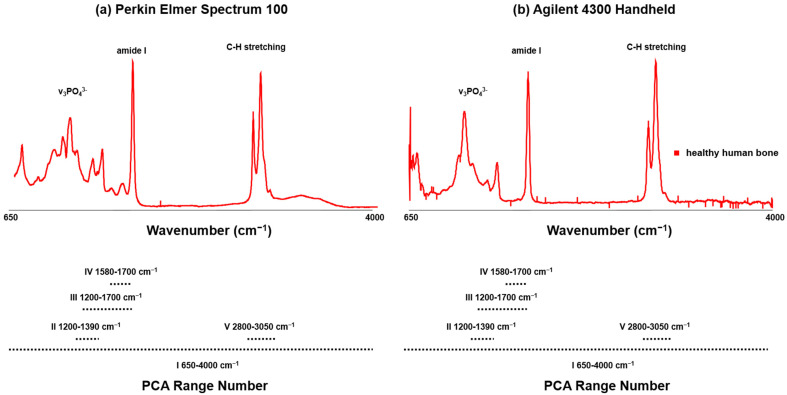
Representative MIR (**a**) Perkin Elmer Spectrum 100 and (**b**) Agilent 4300 Handheld instrument spectrum of a non-infected bone sample. The most important bands are emphasized. Range numbers for PCA analysis are indicated with dashed lines.

**Figure 3 bioengineering-10-01018-f003:**
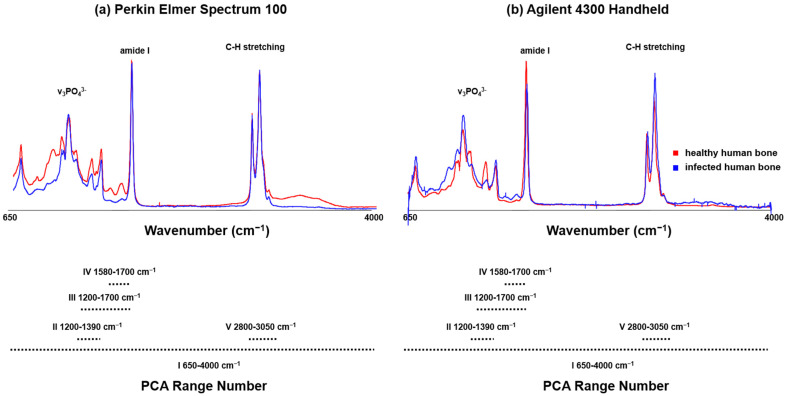
Representative MIR (**a**) Perkin Elmer Spectrum 100 and (**b**) Agilent 4300 Handheld instrument spectrum of non-infected (red) and infected bone sample (blue) are presented. The mineral and matrix bands are emphasized. Range numbers for PCA analysis are indicated with dashed lines.

**Figure 4 bioengineering-10-01018-f004:**
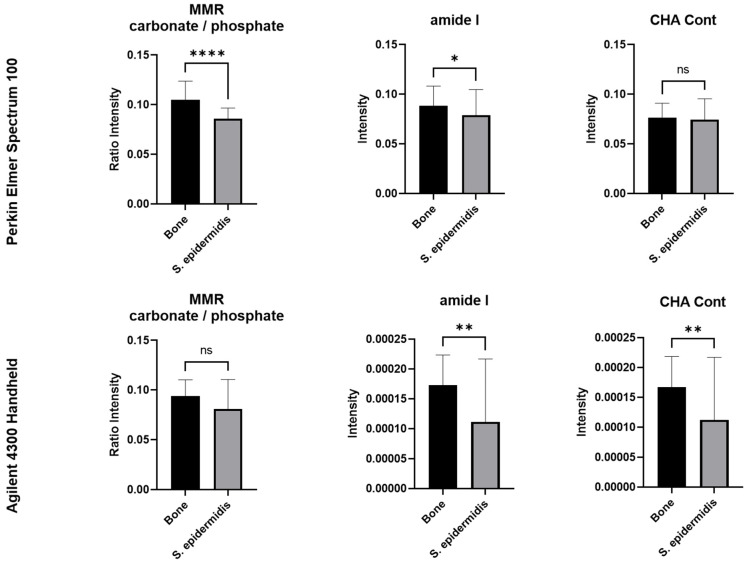
MIR-derived Agilent 4300 Handheld and Perkin Elmer Spectrum 100 instrument spectral markers. * Significant; ** high significant; **** highly significant difference between means; ns, not significant.

**Figure 5 bioengineering-10-01018-f005:**
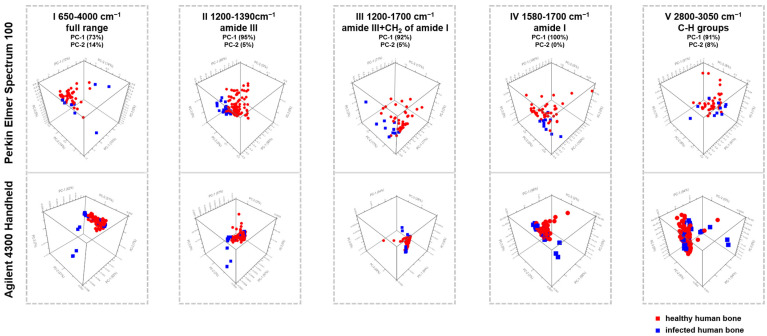
Three-dimensional score plots comparing the MIR Agilent 4300 Handheld and Perkin Elmer Spectrum 100 spectra. The plots are labeled (**I**–**V**) and show the scores between PC1 and PC2 for non-infected versus infected bone samples. Range numbers with the corresponding assignment and spectral region are indicated with a dashed border.

**Table 1 bioengineering-10-01018-t001:** Sample characteristics.

Age (Years)	Total Number with Gender (W = Female; M = Male)	Inoculation with *Staph. epidermidis* ATCC 12228
>80	W = 1 M = 1	W = 0 M = 0
70–80	W = 9 M = 7	W = 1 M = 2
60–70	W = 4 M = 5	W = 1 M = 0
50–60	W = 6 M = 2	W = 3 M = 0
<50	W = 2 M = 3	W = 2 M = 1

**Table 2 bioengineering-10-01018-t002:** Human-bone MIR spectral markers (I: intensity of the band). The *p*-values < 0.05 are considered significant.

Name	Description	*p*-Values (Two-Sample *t*-Test)	
		Agilent 4300	Perkin Elmer Spectrum 100
Phosphate	ν_3_PO_4_^3−^ Amount of phosphate	0.0021	0.1749
Mineral/matrix (MMR) phosphate/amide I	ν_3_PO_4_^3−^/amide I Mineral component amount to the organic one	0.4428	<0.0001
Mineral quality and crystallinity carbonate/phosphate	ν_1_CO_3_^2−^/ν_1_PO_4_^3−^ Carbonate incorporation extent in the hydroxyapatite lattice	0.0620	<0.0001
Mineral carbonate content (MinCarb)	ν_1_CO_3_^2−^/(C-H) bend; CH_2_ wag	0.8890	<0.0001
Amide I	Amide I of α-helical structures Arrangement and quantity of collagen	0.0032	0.0371
CH-aliphatic content (CHACont)	CH_2_ stretching	0.0080	0.6949

**Table 3 bioengineering-10-01018-t003:** Principal component analyses (PCAs) for 40 non-infected vs. 10 infected human bone samples measured with Agilent 4300 Handheld and Perkin Elmer Spectrum 100 were compared. Figure 4 displays the PCA plots.

Wave Number Range Number	PCA Perkin Elmer Spectrum 100	PCA Agilent 4300 Handheld	Assignment	Spectral Region
I	PC-1 (73%) PC-2 (14%)	PC-1 (62%) PC-2 (21%)	Full wavenumber range	650–4000 cm^−1^
II	PC-1 (95%) PC-2 (5%)	PC-1 (97%) PC-2 (2%)	Amide III	1200–1390 cm^−1^
III	PC-1 (92%) PC-2 (5%)	PC-1 (64%) PC-2 (28%)	Amide III, CH_2_ deformation (wagging) of protein, amide I	1200–1700 cm^−1^
IV	PC-1 (100%) PC-2 (0%)	PC-1 (98%) PC-2 (2%)	Amide I	1610–1700 cm^−1^
V	PC-1 (91%) PC-2 (8%)	PC-1 (91%) PC-2 (8%)	C-H groups (bending and stretching modes)	2800–3050 cm^−1^

## Data Availability

The data presented in this study are available upon request from the corresponding author.

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
