# Peer review of "Comparison of Mid-Infrared Handheld and Benchtop Spectrometers to Detect Staphylococcus epidermidis in Bone Grafts"

_bioengineering, 2023, doi:10.3390/bioengineering10091018_

Round 1

Reviewer 1 Report

Comments

It is an interesting paper, can be published after a minor revision as suggested below.

1.There are some typo errors, for examples, see below lines, Captial letter is coming in the middle of the sentence.

In section 2.2, ……We carefully incubated The broth were

Section 2.5,….. and The

2.In Figs. 2-3, both left and wright figures, x-axis title has mistaken, w is missing in left side, in both cases, w should be capital letter

3.In Figs.2-3, author can add (a) and (b) for both figure and mention it in the caption with more clarity for readers.

4.Authors did not explain, what is the detection limit’s of the (Agilent 4300 Handheld) and (Perkin Elmer Spectrum 100) for Staphylococcus epidermidis?

5.Are these devices can only show the bone defect? Or Are these devices can detect Staphylococcus epidermidis? These are not explained clearly.

6.After how many days of infections, these devices can start to show signal for infection.

7.What is the effect of moisture or humidity of the sample on the spectrum recording?

Author should address these questions carefully.

Author Response

Dear Editor, Dear Reviewers,

Thank you for considering this paper and for the constructive comments. We changed the paper according to the comments made by the reviewers. In this response letter, we will document all our answers to the reviewers and state where the changes were applied in the final manuscript:

Rev 1

Comments

It is an interesting paper, can be published after a minor revision as suggested below.

1.There are some typo errors, for examples, see below lines, Captial letter is coming in the middle of the sentence.

In section 2.2, ……We carefully incubated The broth were

AW: The spelling mistakes were corrected according to the reviewer’s comment.

Section 2.5,….. and The

AW: The spelling mistakes were corrected according to the reviewer’s comment.

2.In Figs. 2-3, both left and wright figures, x-axis title has mistaken, w is missing in left side, in both cases, w should be capital letter

AW: We have corrected the spelling mistakes and capitalized “w”.

3.In Figs.2-3, author can add (a) and (b) for both figure and mention it in the caption with more clarity for readers.

AW: Accordingly we added (a) and (b) for both figures and mentioned them in the legend.

4.Authors did not explain, what is the detection limit’s of the (Agilent 4300 Handheld) and (Perkin Elmer Spectrum 100) for Staphylococcus epidermidis?

AW: The pathogen's detection limit (LOD) was not determined in this pilot study since we incubated the bone fragments only for 48 h. This information is now added into the discussion: “Therefore, the pathogen's detection limit (LOD) was not determined. This should be done in a further study by calibration standards with known pathogen concentrations assisting in LOD determination.”

5.Are these devices can only show the bone defect? Or Are these devices can detect Staphylococcus epidermidis? These are not explained clearly.

AW: We added the following text in the discussion: “These results using MIR spectroscopy are in line with previously published Raman's investigations [43,91]. Such direct detection of bone infection via MIR or Raman spectroscopy is challenging due to the complexity of bone tissue, low pathogen concentrations, and spectral overlaps. Nevertheless, with the integration of complementary techniques and careful optimization of sample preparation and data analysis, there is potential for MIR spectroscopy to contribute to Laser-independent detection and characterization of bone infections.“

6.After how many days of infections, these devices can start to show signal for infection.

AW: For this pilot study, only a 48-hour incubation time was tested (see MM section 2.2. Development of Biofilm on Bone Allografts). Based on the results, it can be concluded that after 48 hours of incubation, both the Agilent 4300 Handheld FT-IR and the Perkin Elmer Spectrum 100 ATR infrared spectroscopy benchtop instrument detected bone infection. These devices were able to detect the presence of infection within the timeframe of 48 hours. Thus the study's objective was achieved to compare the effectiveness of the two devices. A separate study examining various incubation times will be necessary to understand the devices' detection abilities comprehensively. We added this information also in the discussion.

7.What is the effect of moisture or humidity of the sample on the spectrum recording?

AW: Bones were properly dried before analysis to minimize the influence of external moisture (missing information was added to the MM section). Also, the ATR measurements were conducted in a controlled environment with controlled humidity levels and temperature (missing information was added to the MM section).

Author should address these questions carefully.

With thanks and kind regards

Reviewer 2 Report

The authors carried out a study to diagnose Staphylococcus epidermidis in human bone grafts, making a comparison between
an Agilent 4300 Handheld Fourier-transform infrared with the Perkin
Elmer Spectrum 100 attenuated-total-reflectance infrared spectroscopy benchtop instrument on 40 non-infected and ten infected human bone samples with Staphylococcus epidermidis.

The study is interesting, well organized and the the findings are supoorted by the results. The limit of such study is the small numr of cases investigated. So I suggest to consider this study as a pilot study and to remark the characteristics of  pilot study in the abstract and in the conclusions, or even in the title.

In addition, some sentences are not clear to me or generate confusion in the reader for example:

1) "Among transplanted tissues, bone follows blood as the most frequently transplanted type" pag 1 line 37

2) "We carefully incubated The broth were incubated at 37°C for 24 hours" pag 3 line 137

After such mandatory changes the paper can be published.

Author Response

Dear Editor, Dear Reviewers,

Thank you for considering this paper and for the constructive comments. We changed the paper according to the comments made by the reviewers. In this response letter, we will document all our answers to the reviewers and state where the changes were applied in the final manuscript:

Rev 2

The authors carried out a study to diagnose Staphylococcus epidermidis in human bone grafts, making a comparison between an Agilent 4300 Handheld Fourier-transform infrared with the Perkin Elmer Spectrum 100 attenuated-total-reflectance infrared spectroscopy benchtop instrument on 40 non-infected and ten infected human bone samples with Staphylococcus epidermidis.

The study is interesting, well organized and the the findings are supoorted by the results. The limit of such study is the small number of cases investigated. So I suggest to consider this study as a pilot study and to remark the characteristics of pilot study in the abstract and in the conclusions, or even in the title.

AW: According to the reviewer’s comment, we considered the study as a pilot study in the abstract, in the conclusions, and in the title.

In addition, some sentences are not clear to me or generate confusion in the reader for example:

1) "Among transplanted tissues, bone follows blood as the most frequently transplanted type" pag 1 line 37

AW: We corrected the misleading sentence according to the reviewer#s comments.

2) "We carefully incubated The broth were incubated at 37°C for 24 hours" pag 3 line 137

AW: We corrected the misleading sentence according to the reviewer#s comments.

After such mandatory changes the paper can be published.

With thanks and kind regards

Reviewer 3 Report

This manuscript is entitled as ‘Comparison of mid-IR handheld and benchtop spectrometers to detect Staphylococcus epidermidis in bone grafts’, but no proper comparison has been performed between the two spectrometers. Not much new information can be found from this manuscript, and those data analysis tools such as Student’s t-test and PCA are just performed without any proper discussion or conclusions. It is not clear either, why the comparison between these two specific spectrometers are important in the first place. Overall, the quality of this manuscript does not seem to have reaced publishable level.

Some minor issues, in case authors would want to improve the manuscript and submit elsewhere:

-      Table 2 : there has to be horizontal lines delineating between independent items. It is very hard to read the table as is.

-      Lines 289-296 : Roman numbers VI and VIII have come out of nowhere.

Author Response

Dear Editor, Dear Reviewers,

Thank you for considering this paper and for the constructive comments. We changed the paper according to the comments made by the reviewers. In this response letter, we will document all our answers to the reviewers and state where the changes were applied in the final manuscript:

Rev 3

This manuscript is entitled as ‘Comparison of mid-IR handheld and benchtop spectrometers to detect Staphylococcus epidermidis in bone grafts’, but no proper comparison has been performed between the two spectrometers. Not much new information can be found from this manuscript, and those data analysis tools such as Student’s t-test and PCA are just performed without any proper discussion or conclusions. It is not clear either, why the comparison between these two specific spectrometers are important in the first place. Overall, the quality of this manuscript does not seem to have reaced publishable level.

AW: This pilot study presents comparable findings of two independent techniques. Data are described under “3.1 Spectroscopy data evaluation”, exemplary findings shown in figures 2-4 and results of statistical analyses shown in figure 4 and table 2. We fully agree to reviewers 1 and 2 that the study should be considered as pilot study, but the objective of this study could be achieved in line with analyses described in current literature (https://doi.org/10.1016/j.saa.2022.121570 and DOI:10.1038/s41598-018-27752-z). The reviewers' suggestions and additional information was integrated into the discussion as suggested.

Some minor issues, in case authors would want to improve the manuscript and submit elsewhere:

-      Table 2 : there has to be horizontal lines delineating between independent items. It is very hard to read the table as is.

AW: Horizontal lines were added as suggested.

-      Lines 289-296 : Roman numbers VI and VIII have come out of nowhere.

AW: We corrected the misleading information.

With thanks and kind regards

Round 2

Reviewer 3 Report

The title clearly says this is about 'comparison' of two MIR spectrometers, but your conclusion is 'both of them are good for bacteria detection, but further studies are needed to assess its merits and drawbacks'. There should have some remarks on the pros and cons of each spectrometer in the Conclusion section, as this was the original point of this paper.

You showed many spectrum graphs and PCA results, but you weren't able to draw useful information from them.

Putting a 'Pilot study' in the title shouldn't be used as a way to get around the problems of poor conclusion. It is not clear what bigger study this work is a 'pilot study' of, when this work is just about comparing two specific products' performance.

Also, You didn't specify why those two commercial products have been chosen. The audience would want to know about general guidance, not the performance of a few specific commercial products.

Author Response

Dear Editor, Dear Reviewers,

Thank you for considering this paper and for the constructive comments. We now changed the paper according to the comments made by the reviewer 3. In this response letter, we will document all our answers to the reviewers and state where the changes were applied in the final manuscript:

The title clearly says this is about 'comparison' of two MIR spectrometers, but your conclusion is 'both of them are good for bacteria detection, but further studies are needed to assess its merits and drawbacks'. There should have some remarks on the pros and cons of each spectrometer in the Conclusion section, as this was the original point of this paper.

AW: We agree, and specified the pros and cons in the introduction and the conclusions.

You showed many spectrum graphs and PCA results, but you weren't able to draw useful information from them.

AW: We still think that the presentation of the spectrum graphs and the PCA results is needed to really show in which spectra and for which graphs the comparability of the data can be expected. We further detailed this in the text.

Putting a 'Pilot study' in the title shouldn't be used as a way to get around the problems of poor conclusion. It is not clear what bigger study this work is a 'pilot study' of, when this work is just about comparing two specific products' performance.

AW: Indeed, the conclusions were adapted as described above. We think that the study has to be extended for there bacteria and fungi in a larger sample size, therefore we consider this study as a „pilot study“.

Also, You didn't specify why those two commercial products have been chosen. The audience would want to know about general guidance, not the performance of a few specific commercial products.

AW: The reason for choosing these two instruments was the size of these instruments. This was now explained in the introduction, together with the question whether this small instrument could provide comparable data. Thank you for your feedback on this issue, which we now tried to clarify and added into the introduction and the discussion sections.

With thank the reviewer for their important comments, to clarify the presentation of this work in this manuscript.

With kind regards

Johannes Pallua, corresponding author

Priv.-Doz. MMag.Dr.rer.nat. Johannes Pallua MSc PhD

Univ.-Klinik für Orthopädie und Traumatologie

Anichstraße 35, A-6020 Innsbruck, Austria

Tel.: +43 50 504 80242, Mail: [email protected]